# Future Grain Consumption Trends and Implications on Grain Security in China

Yuanyuan Chen [1,2]  and Changhe Lu [1,2,*]

1   Key Laboratory of Land Surface Pattern and Simulation, Institute of Geographic Sciences and Natural Resources Research, CAS, Beijing 100101, China; chenyy.15b@igsnrr.ac.cn
2   College of Resources and Environment, University of Chinese Academy of Sciences, Beijing 100049, China
*   Correspondence: luch@igsnrr.ac.cn

**Abstract:** Associated with population and income growth, grain consumption in China is expected to increase, and thus has inevitably influenced the food security. Using statistical data of the Food and Agricultural Organization of the United Nations (FAO) from 1978 to 2017, this study adopting the vector auto-regression (VAR) model and implied demand approach, projected the future consumption of major grains (rice, wheat, maize, and soybean) for food, feed, and other uses in China during 2018–2030. On this basis, it further discussed their implications on grain security. The results showed that during 2018–2030, the Chinese dietary structure would continue to shift from food grain to animal foods. As a result, the grain's food consumption will decrease slightly (1.5%), while the feed consumption will increase significantly (31.4%), contributing 71.4% to the total increase of grain consumption. By 2030, the total grain consumption will increase by 20.2% to 846.2 million tons, of which 50.2% will be consumed for feeding animals. In the total consumption, maize will be the largest consumed grain variety, accounting for 39.2%. The security of rice and wheat would be optimistic in the future, while the security of maize and soybeans is likely to decline, and thus needs to be given high priority. These findings have great policy implications for improving the grain security, suggesting that in addition to promote the expansion of maize and soybean growing area by adjusting the cropping structure of the arable land, great efforts should be paid to improve the yield of both crops. In addition, residents should be guided to adjust the dietary structure, and also, it is important to improve the animal feeding efficiency.

**Keywords:** grain consumption; forecast; VAR model; policy implications; China

## 1. Introduction

With the sustained income increase and lifestyle changes since the late 1970s, China's livestock transition has commonly occurred [1]. During 1978–2013, the per capita consumption of beef, mutton, pork, poultry, eggs, milk, and fish increased by 15.8, 8.4, 3.5, 7.7, 4.5, 7.0, and 9.9 times, respectively [2]. Combined with the constant population growth, the total consumption of main livestock and fishery products all increased more than five times [2]. As maize and soybean meals are the main sources of feed [3,4], the rapid increase in the consumption of livestock and fishery products has led to a significant increase (242.5%) in the feed grain consumption [2]. Taking into account food consumption as well as other consumption (processing, seed, loss, and waste) of grain, the total grain consumption increased greatly by 129.3% [2].

Meeting the demand of grain has always been the core of China's agricultural policy [5,6]. Through institutional reform, technology change, gradual market reform and investment increase in agriculture [7–11], China has made remarkable progress in agricultural production. Since 1978, grain production has increased at an annual growth rate of 3.4% and supplied more than 95% of

the domestic consumption (net imports accounted for less than 5% of domestic consumption) until 2007 [12]. After that year, with a rapid increase in the consumption of animal foods and vegetable oils, the grain consumption grew remarkably, and the self-sufficiency declined continuously to 83.7% in 2017 [12,13]. By around 2030, the population is expected to grow to its peak [14]; the income of rural and urban residents will continue to rise, and the urbanization rate will continue to ascend [15]. Under this situation, what changes will take place in grain consumption and what implications can be concluded for future grain security deserve great attention.

Many scholars have used different methods to predict the future grain consumption in China. In general, these predictions were more focused on the food grain demand and didn't sufficiently analyze the feed demand. In addition, some studies were based on data before 2008, and were not able to predict the accelerated increase in grain consumption after that year [16–18]. Some forecasts were just based on the Chinese household consumption survey data that didn't include the off-home consumption, and thus the results were obviously underestimated [19–21]. Many projections were not specified to the grain variety or consumption type, and thus had limited policy significance. For these reasons, and to provide more implications for sustainable grain security, this study conducted a predictive analysis on the different types of consumption (food, feed, and others) of grain and major grain varieties (rice, wheat, maize, and soybean) during 2018–2030. Other cereal, pulses, and potatoes were also taken into account, but in the analysis, only the main grain varieties and the total grain were focused on. The food consumption was projected with the vector auto-regression (VAR) model using the statistical data of Food and Agriculture Organization (FAO), feed consumption was estimated with the implied demand approach, and other consumption was calculated as a percentage of the sum of food and feed consumption. Based on the forecast results and prospects of the grain production, the grain security level was analyzed. At last, some policy suggestions for improving the grain security were proposed.

## 2. Data and Methods

### 2.1. Data

Two major sources of China's consumption data are available: the household consumption survey data and FAO food balance sheet data. The household survey data that were carried out by the National Bureau of Statistics of China (NBSC) included only "at-home" consumptions of various food types for urban and rural residents in China, but excluded the "away-from-home" consumptions (e.g., restaurants, guest consumption, and other food service outlets) [22], so the records were smaller than actual food consumption [23,24]. The FAO food balance sheet included the two consumptions, as the consumed amount of various food types was calculated by subtracting feed, seed, processing, losses, and wastes from the total supply, according to the principle of supply–demand balance [25]. For this reason, this study used FAO data for the analysis. As the FAO food balance sheet was only updated to 2013, the per-capita consumption of various food types in 2014–2017 was estimated using the household survey records, taking into account the excluded "away-from-home" consumption. This part accounted for 40%, 67%, 63%, 53%, 44%, 57%, 68%, and 70% of the total for grain, beef, mutton, pork, poultry, eggs, milk, and fish, respectively, which was calculated by comparing the two sources of data in 2013. In order to maintain consistent statistical caliber, the harvested area and yield of different grain crops used the data from the FAO Statistical Database. Income data and the urbanization rate were collected from the China National Statistical Database. Predicted population data for 2018–2030 were derived from the medium scenario of the United Nations population projection [14].

## 2.2. Methods

### 2.2.1. VAR Model Building and Testing

Food grain consumption was projected with the vector auto-regression (VAR) model. VAR is an important technique for modeling multivariate time series, and has been widely used in a variety of applications [26–29]. In the VAR model, the target value of one variable can be regressed and predicted with the lag terms of other variables [28]. Changes in food consumption in China are often explained by changes in real income and urbanization level [30–32]. Based on the data of per-capita consumption of different grain varieties and animal foods, per-capita income (at constant prices in 1978), and urbanization rate during 1978–2017, the VAR model was constructed [28]:

$$Y_t = A_1 Y_{t-1} + A_2 Y_{t-2} + \cdots + A_p Y_{t-p} + \mu t \tag{1}$$

In the model, $Y_t$ is a three-dimensional variable vector, namely, $Y_t = \begin{bmatrix} Y_{1ti} \\ Y_{2t} \\ Y_{3t} \end{bmatrix}$, $Y_{1ti}$, $Y_{2t}$, and $Y_{3t}$ is the time-series data of per-capita consumption of food$_i$, per capita income (at constant prices in 1978), and urbanization rate, respectively. p is the lag length and $t$ is the number of samples. $A_1$, $A_2$, and $A_p$ are the coefficient matrix to be estimated. $\mu t$ is a k-dimensional disturbance vector. We used Eviews 8.0 software to build the VAR models in this study. The modeling process is as follows.

In the first step, we examined the stability of the data. To establish a VAR model, the time-series data should be stationary; otherwise, the problem of spurious regression may occur. The augmented Dickey–Fuller (ADF) unit root test method was used to test the stability of the per-capita consumption of various foods, per-capita real income, and urbanization rate during 1978–2012. The test results showed that they were non-stationary at the 5% significance level, and thus we conducted difference processing to make the data stationary (Appendix A, Table A1).

In the second step, the optimal lag order *p* of the model was determined according to the judgment criteria, including the final prediction error (FPE), Akaike information criterion (AIC), Schwarz criterion (SC), and Hannan–Quinn (HQ) criterion. As shown in Appendix A, Table A2, when each model has first-order lag, the values of these criteria are less than those that have second-order lag, indicating that the optimal lag order for each model should be 1.

In the third step, after determining the lag order, the VAR models were constructed based on Formula (1). As shown in Appendix A, Table A3, the change in per-capita consumption of food$_i$ in year *t* was largely explained by the changes in per-capita real income, urbanization rate, and per-capita consumption of food$_i$ in year *t-1*.

The fourth step is to test the stability of the established model. If the absolute value of the inverse root of the model's characteristic equation is less than 1, the model is stable; otherwise, it is unstable [28]. The test results showed that the inverse roots of the characteristic equation of the models were all located in the unit circle (Appendix A, Figure A1); i.e., the absolute values of inverse roots were less than 1, implying that each established VAR model was stable.

In the fifth step, the per-capita consumption of various foods in 2013–2017 was predicted based on the established VAR models, and the forecast values were compared with the statistical values. The results showed that the prediction accuracy of each VAR model was above 94% (Appendix A, Table A4), indicating that the models can be used to predict the future per-capita food consumption.

### 2.2.2. Future Consumption Predicting

Before forecasting, the per-capita real income and urbanization rate during 2018–2030 need to be assumed. During 1978–2017, the per-capita real income and urbanization rate kept a quick growth, from 171.2 to 4071.8 CNY (Chinese Yuan). However, the growth rate has been declining from 10.4% to 7.3% since 2010. This slowing down in the growth is expected in the near future, not only for the

income, but also for the urbanization rate [3,6]. Thus, we assumed that the average annual growth rate of per-capita real income will be 7.0% in 2018–2020, 6.5% in 2021–2025, and 6.0% in 2026–2030, while the urbanization rate will increase by 1.1% annually in 2018–2020, 0.9% in 2021–2025, and 0.7% in 2026–2030. These assumptions were input to the established VAR models to predict the per-capita consumption of grain varieties and animal foods during 2018–2030. Further, combined with the United Nations population projection, the total consumption of food grain and animal foods was estimated.

The consumption of feed grain was estimated with the implied demand approach. In this approach, demand for feed grain was calculated by multiplying the outputs of animal foods by the feed–meat conversion ratios [33]. Referring to the existing studies [21,34], the feed–meat conversion ratios (the amount of grain consumed in the production of per kilogram of animal products) in Table 1 were used to calculate the grain consumption for feeds.

**Table 1.** Feed–meat conversion ratios (kg/kg) of major animal foods in China.

| Animal Foods | Pork | Beef and Mutton | Poultry | Eggs | Milk | Fish * |
|---|---|---|---|---|---|---|
| Conversion ratios | 2.8 | 1.0 | 2.0 | 1.7 | 0.3 | 0.9 |

\* Fish in this paper refers to fishery products.

During 2003–2013, the grain consumption for processing, seeds, and wastes, which is termed as other consumption in this study, occupied approximately 20–25% of the total. For simplicity, we assumed that the proportion is 23%, which was used to estimate the amount of other consumption. By summing up the food, feed, and other consumption, the total consumption of grain and different varieties was obtained.

## 3. Results

### 3.1. Changes in Per-Capita Consumption of Grain Varieties and Animal Foods

During 2018–2030, the total food grain consumption per capita was predicted to decline from 163.5 kg in 2017 to 161.3 kg in 2020, 158.9 kg in 2025 and 157.1 kg in 2030 (Figure 1a). It showed a similar trend for rice, wheat, maize, and soybeans, with the per-capita consumption decreasing from 78.0, 63.6, 6.5, and 7.6 kg in 2017 to 74.9, 62.4, 5.8, and 7.0 kg in 2030, respectively. The decrease rate will be slowing down: the per-capita food grain consumption was predicted to decrease by 0.5% per year during 2018–2020, by 0.3% in 2021–2025, and by 0.2% in 2026–2030.

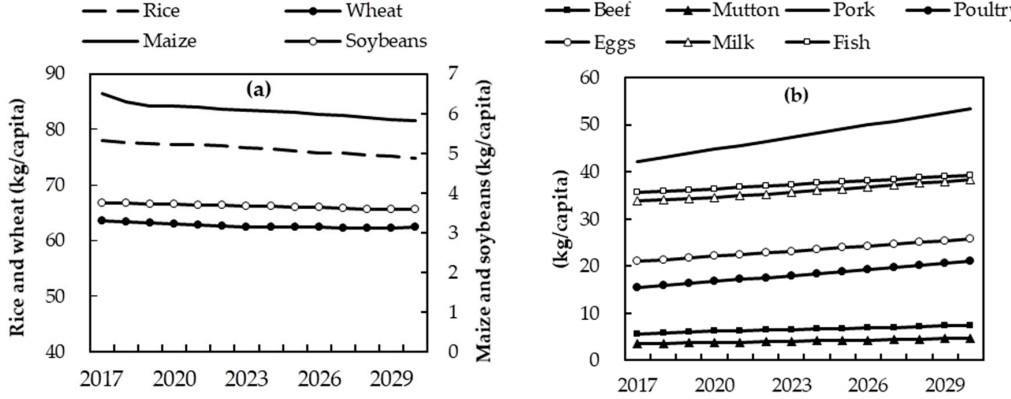

**Figure 1.** Per-capita consumption of main grain varieties (**a**) and animal foods (**b**) during 2017–2030.

By contrast, the per-capita consumption of animal foods will increase significantly, especially for beef, mutton, and poultry meats (Figure 1b). By 2030, the per-capita consumption of beef, mutton, pork, poultry, eggs, milk, and fish will reach 7.4, 4.7, 53.5, 21.1, 25.8, 38.3, and 39.3 kg/capita, respectively,

which is 30.5%, 34.8%, 26.7%, 35.2%, 22.9%, 13.1%, and 10.2% higher than that in 2017, respectively. Similarly, the growth rates of the per-capita consumption of main animal foods will decline generally in the future. For example, the per-capita consumption of beef, mutton, and poultry will grow at an average annual rate of 2.7%, 2.8%, and 2.6% in 2018–2020, but will fall to 2.0%, 2.3%, and 2.4% in 2021–2025, and 1.8%, 2.1%, and 2.2% in 2026–2030, respectively.

### 3.2. Changes in Different Types of Consumption of Grain and Main Varieties

During the same period, the population would increase 2.3% to 1441.2 million. This population growth speed was lower than the decline rate of per-capita food grain consumption; as a result, the total food grain consumption will decrease 1.5%, from 230.1 million tons (Mt) to 226.4 Mt (Table 2). Accordingly, its proportion in the total grain consumption will fall from 32.7% to 26.8%. However, the total grain consumption for feeding animals will greatly increase by 31.4%, from 323.3 Mt to 424.8 Mt. With this quick increase, the proportion of feed grains in the total grain consumption will ascend from 45.9% to 50.2%. Meanwhile, the other grain consumption will also increase greatly (29.3%) from 150.7 Mt to 195.0 Mt. Taken those together, the total grain consumption was predicted to increase by 20.2% from 704.3 Mt to 846.2 Mt, which is largely (71.4%) contributed by the increased feed consumption. As the growth rate of population and per-capita consumption of animal foods were predicted to decline, the annual growth rate of total grain consumption will also decline gradually, from 2.6% in 2018–2020 to 1.3% in 2021–2025, and 1.0% in 2026–2030.

**Table 2.** Consumption of main grain products in 2017, 2020, 2025, and 2030 (Mt).

| Consumption | Rice | Wheat | Maize | Soybeans | Grains |
|---|---|---|---|---|---|
| | | 2017 | | | |
| Food | 110.0 | 89.6 | 9.2 | 5.3 | 230.1 |
| Feed | 22.5 | 38.8 | 218.4 | 19.3 | 323.4 |
| Others | 13.3 | 12.8 | 34.2 | 78.8 | 150.8 |
| Total | 147.4 | 141.3 | 261.8 | 103.4 | 704.3 |
| | | 2020 | | | |
| Food | 110.1 | 89.8 | 8.8 | 5.3 | 229.8 |
| Feed | 24.5 | 40.3 | 235.4 | 26.2 | 356.0 |
| Others | 13.5 | 13.0 | 36.6 | 97.6 | 173.2 |
| Total | 148.0 | 143.1 | 280.9 | 129.1 | 759.1 |
| | | 2025 | | | |
| Food | 109.5 | 89.8 | 8.7 | 5.3 | 228.6 |
| Feed | 27.3 | 44.3 | 259.0 | 29.7 | 391.7 |
| Others | 13.7 | 13.4 | 40.2 | 108.3 | 182.0 |
| Total | 150.5 | 147.5 | 307.8 | 143.2 | 802.4 |
| | | 2030 | | | |
| Food | 108.0 | 90.0 | 8.4 | 5.2 | 226.4 |
| Feed | 31.4 | 48.0 | 280.9 | 32.9 | 424.8 |
| Others | 13.9 | 13.8 | 43.4 | 117.9 | 195.0 |
| Total | 153.3 | 151.8 | 332.7 | 155.9 | 846.2 |

The consumption structure of main grain varieties will be changed accordingly. For the staple food grains of rice and wheat, the consumption for food will maintain relatively stable, while the consumption for feed and others will increase slightly by 5.9 Mt (4.0%) and 10.5 Mt (7.4%), respectively. Furthermore, the proportion of rice and wheat to the total grain consumption will decline from 20.9% and 20.1% to 18.1% and 17.9%, respectively. As the most important feed source, maize consumption for feed will increase significantly by 62.6 Mt (28.7%), while the consumption for food will decline a little bit, and the consumption for other purposes will increase modestly (9.2 Mt). As a result, the total consumption of maize will increase greatly to 332.7 Mt, accounting for 39.2% of total grain consumption. Of the total maize consumption increment, 88.0% was predicted to be consumed for feed. Soybean is the main source of protein feed and an important raw material for oil extraction. Its consumption

for food will remain stable, while the consumption for feed and others (mainly oil processing) will increase significantly by 13.5 Mt and 41.6 Mt, contributing 24% and 76% of its total consumption increment, respectively.

## 4. Discussion

### 4.1. Implications on the Grain Security

In 2017, the production of rice, wheat, maize, soybeans, and total grain was 143.4, 134.3, 259.1, 13.3, and 589.2 Mt, respectively. This implies that to meet the demand in 2030, the production of rice, wheat, maize, and soybeans should increase by 6.9%, 13%, 28.4%, and 10.7 times, respectively, while the total grain production should increase by 43.6%. As China's arable land resources have been mostly exploited, to promote the growth of grain production, emphasis should be on adjusting the cropping structure of the existing arable lands and improving the crop yield.

During 1978–1998, the harvest area of rice and wheat changed slightly. Then, they declined obviously between 1999–2003 due to the government's relaxation of grain production and the implementation of the Grain-for-Green Program. After the following five years of continuous expansion, the harvest areas of rice and wheat have remained stable. The harvest area of maize and soybean increased slightly before 2003. However, as a consequence of the increasing demand for feed grain, the harvest area of maize expanded sharply, with an average annual growth rate of 5.5% during 2004–2015. Over the same period, the harvest area of soybean decreased at an average annual rate of 2.9% due to lower yield and benefits. Since 2015, China has begun to support the production of soybean. In 2015–2017, the harvest area of soybean increased by 0.8 million hectares (Mha), while the acreage of maize reduced by 2.57 Mha (Figure 2a).

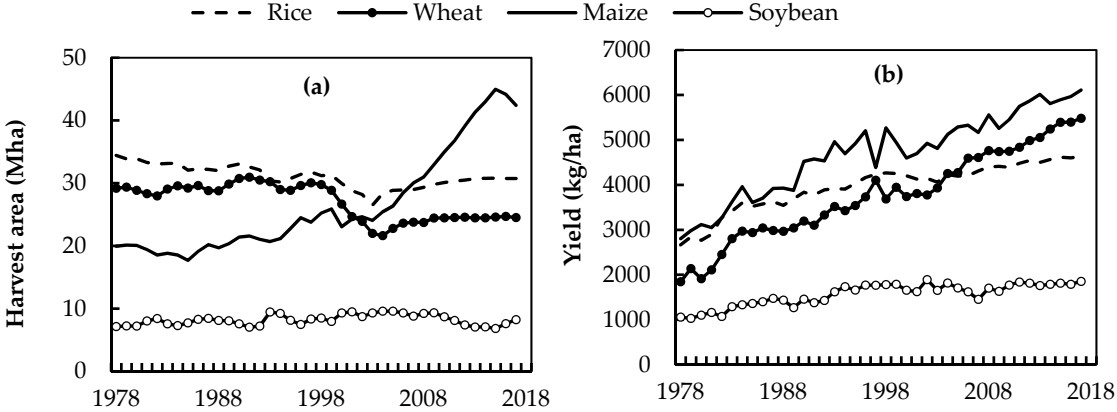

**Figure 2.** Harvest area (**a**) and yield (**b**) of main grain crops during 1978–2017.

In the near future, with the food grain consumption gradually stabilized, the harvest area of rice and wheat is likely to remain at 31.0 and 25.4 Mha, respectively. With the increasing feed demand, especially protein feed [3], the soybean revitalization plan is likely to be maintained. As a result, the harvest area of soybean is likely to continue to expand at the current rate of 3%. By 2030, it will reach 10.8 Mha, which is 46.9% higher than that in 2017. The expansion of soybean cultivation will compress the acreage of maize, so the maize acreage will continue to decline for some time. However, as the most important source of feed, the harvest area of maize will not decline constantly, and is likely to gradually recover after 2020 [35]. However, the recovery of maize harvest area is difficult to predict, as it was heavily affected by agricultural policies.

During 1978–2017, the yield of rice, wheat, maize, and soybean has been improved steadily with an annual growth rate of 1.5%, 3.0%, 2.2%, and 1.6%, and reached 4620, 5402, 5942, and 1816 kg/ha in 2017, respectively (Figure 2b). The quick improvement in yield was mainly due to the progress of agricultural science and technology. Hybrid rice has been developed by China's scientists in the late 1970s, and technological innovations in wheat and maize have also been significant [7]. The contribution of agricultural scientific and technological progress to agricultural production has exceeded 56% [12]. The increase in agricultural investment was another important driver. For example, the irrigated arable land area has expanded by 50.8%, and the application of chemical fertilizers and pesticides has increased by 5.6 and 1.3 times, respectively [12].

In the future, the application of chemical fertilizers and pesticides per hectare of arable land may not grow much, as their levels were 3.4 and 3.8 times the world average respectively, and the negative impact on the ecological environment and human health was gradually prominent [36,37]. However, the irrigation area was expected to expand constantly, as a total of 67.2 Mha of farmland (50% of total) still lacks effective irrigation [12]. Besides, the Chinese government clearly stated an intention to "vigorously improve agricultural science and technology, and strengthen the promotion of advanced and applicable technologies". With the strong support, the agricultural science and technology progress is very likely to continue, and thus drive the grain yield to improve. As a result, the yield of grain crops would continue to improve. Presently, the yield of rice and wheat is 50% and 55% higher than the world average [38], and the yield gap (difference between actual and potential yield) has reduced to 29.5% and 25.4%, respectively [39]. Therefore, the yield growth of rice and wheat may be slowing down. However, the growth potential of maize and soybean is high, with yield gaps of 52.4% and 63.9%, respectively [39], so their yields are likely to maintain the current growth trends. However, the literature data showed that the future climate change would probably have a negative impact on the yield of grain crops [40,41], and thus, the yield growth may be slightly slowed by climate warming and meteorological hazards.

Based on the above analysis, if the yield of rice and wheat remain in line with the growth trend, their total production will increase to 154.2 and 156.6 Mt by 2030, exceeding the demand by 0.6% and 3.2%, respectively. If the yield growth rate of rice and wheat declined by 50% (0.3% and 0.8% annually respectively during 2018–2030), their self-sufficiency ratio can still exceed 97%. That is, the security prospect of rice and wheat is optimistic. For maize, its yield is likely to improve to 7006 kg/ha by 2030. If the harvest area recovered to 45.0 Mha (as in 2015), the total production will increase to 315.3 Mt, and the self-sufficiency ratio could be slightly decreased to 94.8%. For soybean, the yield is likely to improve to 2115 kg/ha in 2030 and the total production would increase by 71% to 22.8 Mt. Even so, the production gap will continue to widen to 133.1 Mt. In other words, the future security level of maize and soybean is likely to decline, and needs to be given high priority. According to the fact that the production of these four varieties accounts for 93% of the total grain production, the total grain production will increase to 678.7–695.1 Mt, and can only meet 80.3–82.1% of the total demand in 2030.

*4.2. Policy Suggestions*

According to the above analysis, meeting the demand of maize and soybean is the key to achieving grain security. On one hand, measures should be taken to improve the production capacity of maize, and especially soybeans. Firstly, over the past decades, many areas, including non-dominant areas of maize, have expanded maize cultivation. This not only took up the acreage of other crops, but also lowered the overall yield of maize. The cultivation of maize in disadvantageous areas should be reduced appropriately, while it in areas where it is advantageous, it should be consolidated and enhanced. The planting of maize for grain should be reduced and the planting of silage maize should be expanded. Secondly, the cultivation of soybean should be expanded continuously. There may be feasible ways to increase the planting of soybeans in the Northeast Plain, and promote the rotation and intercropping of maize and soybeans in the Huang-Huai-Hai Plain. Thirdly, the yield of both should be

promoted. To this end, investments in agricultural science and technology innovation systems should be increased, especially in research on high-yield varieties of soybeans.

On the other hand, reducing the consumption of maize and soybeans is another effective way. Firstly, adjusting the consumption structure of livestock products should be encouraged to reduce the feed demand. Considering the eating habits of Chinese residents, reducing the proportion of pork in the total meat consumption and increasing the proportion of poultry could be a feasible way. In addition, residents should be encouraged to increase the consumption of milk and dairy products. Secondly, improving the efficiency of feed usage was also an important solution. It is estimated that if the feed–meat conversion ratios of major livestock products were decreased by 0.1 by rising feed usage efficiency, the consumption of maize and soybeans will decrease by 20.8 and 12.9 Mt, respectively. To improve the feed utilization, it is necessary to popularize the excellent breeds of livestock, develop new and efficient breeding models, and optimize and promote feed formulation technology.

*4.3. Comparison with Other Studies*

As mentioned in the introduction, many scholars and organizations have predicted China's future grain consumption prior to this study, including the Chinese Academy of Agricultural Sciences (CAAS) [35], the Organization for Economic Cooperation and Development and Food and Agricultural Organization of the United Nations (OECD-FAO) [42], and the United States Department of Agriculture (USDA) [43]. These studies are relatively up-to-date and systematic. They adopted a partial equilibrium model to predict the different types of consumption of various foods for the following 10 years, i.e., 2018–2027.

In comparison, we found that the predicted results for the feed consumption of the main varieties in these three studies were less than our projections, but their forecasts for food grain consumption were consistent with ours. For example, our and these studies all predicted that the consumption of rice will be 143.3–152.1 Mt in 2025, while our forecast for maize consumption was about 50 Mt higher than the others (Table 3). The consistency of food grain forecasting suggests that it is appropriate to predict the level of food consumption with a VAR model. The difference in the prediction of feed grain consumption may be due to the different estimation methods of feed consumption. Although the specific forecast values are different, our and these studies all predicted that the consumption of food grains (mainly rice and wheat) will not change much, while the consumption of feed grains (mainly maize and soybean) will increase significantly in the following decade.

**Table 3.** Consumption forecasts of grain and main varieties in other studies in 2025 (Mt).

| Studies | Rice | Wheat | Maize | Soybean | Grain |
|---------|------|-------|-------|---------|-------|
| This study | 150.5 | 147.5 | 307.8 | 143.2 | 802.4 |
| CAAS | 152.1 | 132.7 | 227.0 | | |
| OECD-FAO | 148.9 | 136.1 | 250.7 | 128.0 | 763.8 * |
| USDA | 143.3 | 134.4 | 270.7 | 134.8 | 776.7 * |

* Calculated based on the proportion of the sum of rice, wheat, maize, and soybeans consumption to total grain consumption. CAAS: Chinese Academy of Agricultural Sciences, OECD-FAO: Organization for Economic Cooperation and Development and Food and Agricultural Organization of the United Nations, USDA: United States Department of Agriculture.

## 5. Conclusions

During 2018–2030, the diet structure will continue to shift from grain foods to animal foods, even though the change rate will slow down gradually. As a result, the consumption of rice and wheat will not increase much, while the consumption of maize and soybeans will increase significantly. The food grain consumption will decline slightly, while feed grain consumption will increase significantly, which is the main contributor to the total increase of grain consumption. By 2030, the total grain consumption will reach 864.2 Mt, of which feed is the largest type of consumption, accounting for 50.2%, and maize is the most consumed variety, accounting for 39.2%. With the crop yield improvement and possible

changes in cropping structure in the future, rice and wheat are likely to be nearly self-sufficient, while the production gap of maize and soybeans is likely to continue to widen. That is, the security prospect of rice and wheat is optimistic, while the future security level of maize and soybeans is likely to decline, and needs to be given high priority in the future.

The findings of this study provide direction for ensuring the grain security in China. To improve the grain security level, meeting the demand of maize and soybean is the key. On one hand, the production of maize and soybeans should be enhanced. To this end, it is suggested that it should adjust the cultivation of maize, expand the cultivation of soybeans continuously, and promote the yield of both. On the other hand, the consumption of maize and soybeans should be reduced by adjusting the residents' dietary structure and improving the efficiency of feed usage. Our findings also suggest that the research on the production potential of maize and soybean needs to be strengthened in the future.

**Author Contributions:** Conceptualization, C.L. and Y.C.; methodology, Y.C.; software, Y.C.; validation, C.L.; formal analysis, Y.C.; investigation, C.L.; resources, C.L.; data curation, Y.C.; writing—original draft preparation, Y.C.; writing—review and editing, C.L.; visualization, Y.C.; supervision, C.L.; project administration, C.L.; funding acquisition, C.L.

**Funding:** This research was funded by [The Chinese Academy of Sciences Key Deployment Project] grant number [ZDRW-ZS-2016-6-4-3] and [the National Natural Science Foundation of China] grant number [41671093]. And the APC was funded by [The Chinese Academy of Sciences Key Deployment Project].

**Conflicts of Interest:** The authors declare no conflict of interest.

## Appendix A

**Table A1.** The unit root test results of processed variables. ADF: augmented Dickey–Fuller.

| Variables * | Test (C,T,K) [#] | ADF Statistics | $p$ Value | Significant Level | | | Station-Arity |
|---|---|---|---|---|---|---|---|
| | | | | 1% | 5% | 10% | |
| Δrice | (0,0,0) | −5.208 | 0 | −2.627 | −1.95 | −1.611 | Yes |
| Δwheat | (0,0,0) | −2.157 | 0.032 | −2.627 | −1.95 | −1.611 | Yes |
| Δmaize | (0,0,0) | −6.712 | 0 | −2.627 | −1.95 | −1.611 | Yes |
| Δother cereals | (0,0,0) | −3.843 | 0 | −2.627 | −1.95 | −1.611 | Yes |
| Δsoybeans | (0,0,1) | −6.886 | 0 | −2.627 | −1.95 | −1.611 | Yes |
| Δpulses | (0,0,0) | −6.435 | 0 | −2.627 | −1.95 | −1.611 | Yes |
| Δpotatoes | (0,0,0) | −6.021 | 0 | −2.627 | −1.95 | −1.611 | Yes |
| Δbeef | (C,0,0) | −5.402 | 0 | −3.616 | −2.941 | −2.609 | Yes |
| Δmutton | (C,0,1) | −6.957 | 0 | −3.621 | −2.943 | −2.61 | Yes |
| Δpork | (C,0,0) | −5.237 | 0 | −3.616 | −2.941 | −2.609 | Yes |
| Δpoultry | (C,0,0) | −5.172 | 0 | −3.616 | −2.941 | −2.609 | Yes |
| Δeggs | (C,0,0) | −5.159 | 0 | −3.616 | −2.941 | −2.609 | Yes |
| Δmilk | (C,0,3) | −3.721 | 0.012 | −3.809 | −3.021 | −2.65 | Yes |
| Δfishes | (C,0,0) | −3.909 | 0.005 | −3.616 | −2.941 | −2.609 | Yes |
| ΔΔincome | (0,0,0) | −7.827 | 0 | −2.627 | −1.95 | −1.611 | Yes |
| ΔΔurbanization | (0,0,0) | −10.049 | 0 | −2.627 | −1.95 | −1.611 | Yes |

*Δ food$_i$ represents changes of per-capita consumption of food$_i$. [#] C, T, and K represent the constant term, trend term, and lag length contained in the ADF unit root test, respectively.

**Table A2.** The selection of lag length. AIC: Akaike information criterion.

| Variables | Lag | LogL | LR | FPE | AIC | SC | HQ |
|---|---|---|---|---|---|---|---|
| Rice | 1 | −241.52 | NA | 222.55 * | 13.91 * | 14.31 * | 14.05 * |
| | 2 | −235.37 | 10.24 | 262.94 | 14.07 | 14.86 | 14.35 |
| Wheat | 1 | −215.5 | NA | 74.91 * | 12.82 * | 13.23 * | 12.96 * |
| | 2 | −208.22 | 12.66 | 83.42 | 12.92 | 13.72 | 13.21 |
| Maize | 1 | −123.56 | NA | 11.69 * | 10.97 * | 11.36 * | 11.11 * |
| | 2 | −188.48 | 9.43 | 14.19 | 11.15 | 11.95 | 11.44 |
| Other cereals | 1 | −193.26 | NA | 21.02 * | 11.56 * | 11.95 * | 11.69 * |
| | 2 | −191.44 | 3.01 | 31.97 | 11.97 | 12.76 | 12.24 |
| Soybeans | 1 | −188.98 | NA | 12.02 * | 10.99 * | 11.39 * | 11.13 * |
| | 2 | −182.43 | 10.93 | 13.88 | 11.13 | 11.93 | 11.41 |
| Pulses | 1 | −204 | NA | 27.69 * | 11.83 * | 12.23 * | 11.97 * |
| | 2 | −202.32 | 2.3 | 42.62 | 12.26 | 13.05 | 12.53 |
| Potatoes | 1 | −196.63 | NA | 18.38 * | 11.42 * | 11.82 * | 11.56 * |
| | 2 | −189.86 | 11.29 | 20.97 | 11.55 | 12.34 | 11.83 |
| Beef | 1 | −144.12 | NA | 0.99 * | 8.51 * | 8.90 * | 8.64 * |
| | 2 | −137.18 | 11.58 | 1.12 | 8.63 | 9.41 | 8.89 |
| Mutton | 1 | −135.17 | NA | 0.6 * | 8.01 * | 8.41 * | 8.14 * |
| | 2 | −128.8 | 10.61 | 0.71 | 8.16 | 8.94 | 8.43 |
| Pork | 1 | −209.32 | NA | 37.21 * | 12.12 * | 12.52 * | 12.26 * |
| | 2 | −203.2 | 10.21 | 44.01 | 12.29 | 13.08 | 12.56 |
| Poultry | 1 | −183.21 | NA | 11.84 | 10.98 | 11.38 * | 11.12 * |
| | 2 | −173.48 | 16.13 | 11.45 * | 10.94 * | 11.74 | 11.22 |
| Eggs | 1 | −203.55 | NA | 27 * | 11.81 | 12.2 * | 11.94 * |
| | 2 | −194.42 | 15.23 | 27.02 | 11.8 * | 12.59 | 12.07 |
| Milk | 1 | −195.87 | NA | 17.62 * | 11.38 * | 11.78 * | 11.52 * |
| | 2 | −192.2 | 6.11 | 23.89 | 11.67 | 12.47 | 11.95 |
| Fish | 1 | −199.42 | NA | 21.47 * | 11.58 * | 11.97 * | 11.72 * |
| | 2 | −194.71 | 7.86 | 27.46 | 11.82 | 12.61 | 12.09 |

* indicates the lag order selected by the criterion.

**Table A3.** Construction results of vector auto-regression (VAR) models for changes in the per-capita consumption of different kinds of food.

| Predicted Variables | Explanatory Variables * | Coefficients | Standard Deviation | *t*-Value |
|---|---|---|---|---|
| Rice | Rice (-1) | 0.100 | 0.221 | 0.610 |
| | Income (-1) | −0.014 | 0.010 | 1.214 |
| | Urbanization (-1) | 2.320 | 1.061 | 2.186 |
| | C | 0.119 | 0.329 | 0.362 |
| Wheat | Wheat (-1) | 0.749 | 0.090 | 8.020 |
| | Income (-1) | 0.009 | 0.005 | −0.053 |
| | Urbanization (-1) | −0.375 | 0.815 | 0.553 |
| | C | −0.171 | 0.189 | −0.905 |
| Maize | Maize (-1) | −0.057 | 0.198 | −0.331 |
| | Income (-1) | 0.001 | 0.003 | 1.546 |
| | Urbanization (-1) | −0.427 | 0.488 | −2.216 |
| | C | 0.036 | 0.081 | 0.442 |

**Table A3.** *Cont*.

| Predicted Variables | Explanatory Variables * | Coefficients | Standard Deviation | *t*-Value |
|---|---|---|---|---|
| Other cereals | Other Cereals | 0.483 | 0.240 | 1.956 |
| | Income (-1) | −0.002 | 0.003 | −0.454 |
| | Urbanization (-1) | −0.456 | 0.549 | −1.363 |
| Soybeans | Soybeans (-1) | −0.024 | 0.211 | −1.721 |
| | Income (-1) | 0.001 | 0.003 | 0.888 |
| | Urbanization (-1) | 0.127 | 0.511 | 1.393 |
| | C | −0.025 | 0.070 | −0.317 |
| Pulses | Pulses (-1) | −0.024 | 0.219 | −0.412 |
| | Income (-1) | 0.001 | 0.050 | 1.326 |
| | Urbanization (-1) | −0.134 | 0.391 | −0.345 |
| | C | −0.040 | 0.119 | −0.333 |
| Beef | Beef (-1) | 0.384 | 0.125 | 2.351 |
| | Income (-1) | 0.000 | 0.001 | 1.771 |
| | Urbanization (-1) | 0.066 | 0.082 | 1.238 |
| | C | 0.089 | 0.031 | 2.846 |
| Mutton | Mutton (-1) | −0.057 | 0.133 | −0.395 |
| | Income (-1) | 0.001 | 0.001 | 1.393 |
| | Urbanization (-1) | 0.031 | 0.043 | 0.655 |
| | C | 0.081 | 0.022 | 3.593 |
| Pork | Pork (-1) | −0.091 | 0.129 | −0.506 |
| | Income (-1) | 0.004 | 0.005 | 1.244 |
| | Urbanization (-1) | −0.009 | 0.443 | −1.074 |
| | C | 0.873 | 0.183 | 4.764 |
| Poultry | poultry(-1) | 0.116 | 0.162 | 1.421 |
| | income(-1) | 0.004 | 0.162 | 3.269 |
| | urbanization(-1) | 0.129 | 0.002 | 1.064 |
| | C | 0.310 | 0.095 | 3.253 |
| Eggs | Eggs (-1) | −0.180 | 0.147 | 3.932 |
| | Income (-1) | 0.002 | 0.003 | 1.558 |
| | Urbanization (-1) | −0.242 | 0.378 | −1.299 |
| | C | 0.444 | 0.143 | 3.078 |
| Milk | Milk (-1) | 0.817 | 0.249 | 7.210 |
| | Income (-1) | −0.003 | 0.277 | −0.590 |
| | Urbanization (-1) | −0.271 | 0.354 | −0.765 |
| | C | 0.176 | 0.137 | 1.278 |
| Fish | Fishes (-1) | 0.544 | 0.097 | 9.066 |
| | Income (-1) | −0.004 | 0.003 | −0.866 |
| | Urbanization (-1) | −0.195 | 0.479 | −1.046 |
| | C | 0.402 | 0.169 | 2.486 |

* Rice (-1) represents the change in per-capita consumption (kg) of rice in year *t-1*; such representation also applies to other kinds of food. Income (-1) and urbanization (-1) represent the changes in per-capita real income and urbanization rate in year *t-1*, respectively. C represents constant item.

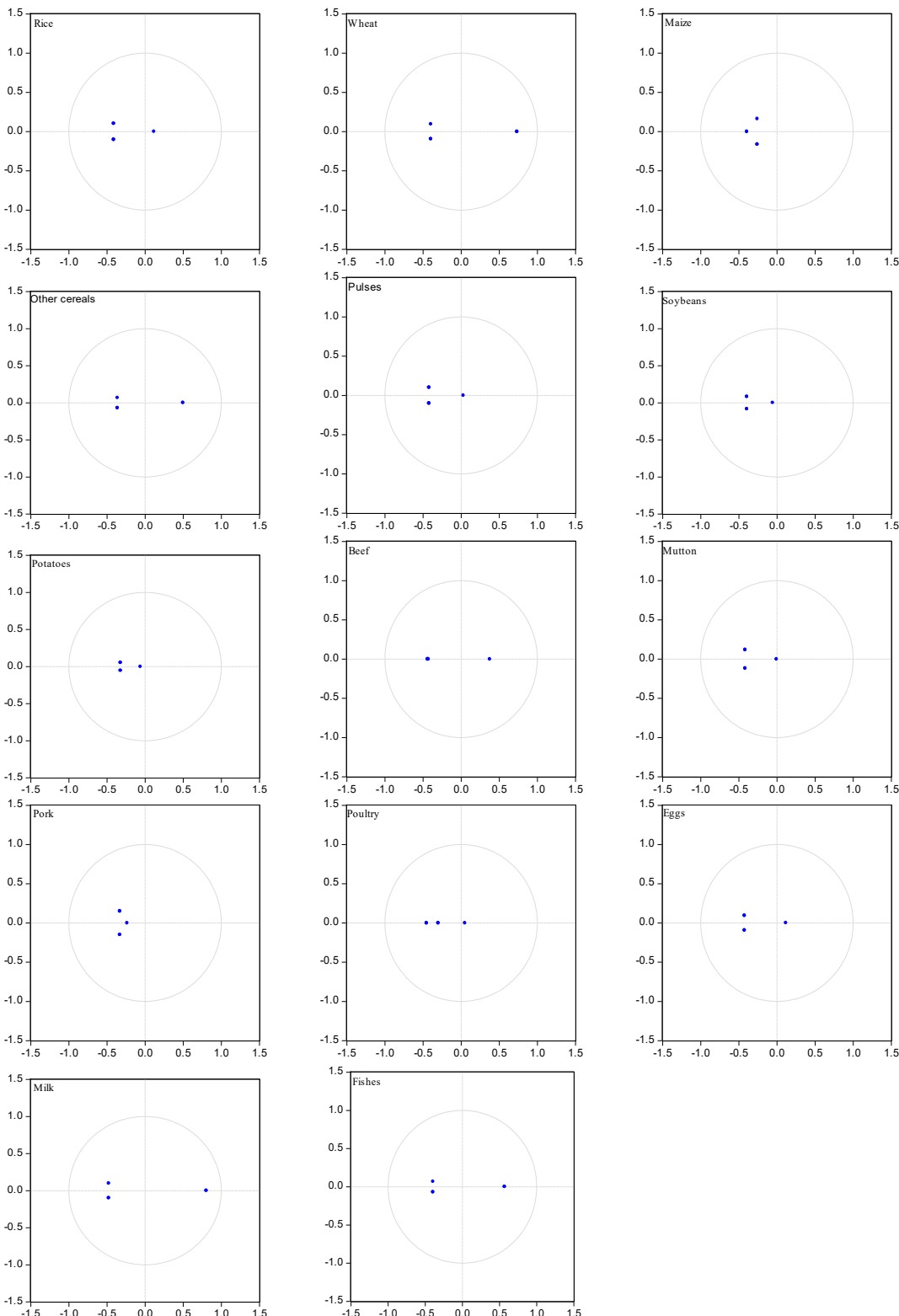

**Figure A1.** Distribution of inverse roots of the characteristic equations of each VAR model.

**Table A4.** The prediction accuracy * (%) of per-capita consumption of various foods by the VAR model in 2013–2017.

| Foods | 2013 | 2014 | 2015 | 2016 | 2017 | Average |
|-------|------|------|------|------|------|---------|
| Rice | 95.3 | 96.6 | 97.1 | 98.5 | 99.9 | 97.5 |
| Wheat | 94.9 | 95.1 | 95.1 | 96.0 | 96.8 | 95.6 |
| Maize | 98.1 | 95.5 | 93.7 | 94.7 | 92.4 | 94.9 |
| Soybeans | 98.0 | 96.8 | 97.1 | 97.9 | 97.5 | 97.5 |
| Grains | 98.9 | 99.3 | 99.8 | 99.9 | 99.6 | 99.5 |
| Beef | 99.7 | 97.1 | 95.8 | 95.1 | 98.2 | 97.2 |
| Mutton | 95.7 | 90.8 | 94.4 | 95.4 | 99.4 | 95.1 |
| Pork | 97.3 | 94.4 | 96.0 | 99.3 | 99.9 | 97.4 |
| Poultry | 96.0 | 96.9 | 98.9 | 95.7 | 99.5 | 97.4 |
| Eggs | 97.7 | 98.4 | 99.8 | 99.4 | 99.8 | 99.0 |
| Milk | 94.2 | 96.4 | 98.2 | 98.9 | 99.5 | 97.5 |
| Fish | 93.5 | 95.2 | 96.4 | 98.0 | 98.8 | 96.4 |

* The model accuracy = $(1 - |estimated\ value - actual\ value|/actual\ value) * 100$.

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
