# Peer review of "Future Grain Consumption Trends and Implications on Grain Security in China"

_sustainability, doi:10.3390/su11195165_

Round 1

Reviewer 1 Report

I think the introduction would benefit from a wider discussion of the evolution of the livestock sector given that feed demand is such a major force. Perhaps you could start with "China’s livestock transition: Driving forces, impacts, and consequences Zhaohai Bai1,2, Wenqi Ma3, Lin Ma1*, Gerard L. Velthof4, Zhibiao. There is also no mention of climate change which could reduce yield growth. 

It would have been good to see some sensitivity analysis regarding key assumptions, e.g.  your assumptions about feed-meat conversions rates are rather pessimistic.

The discussion of policy options should be related to the actions set out in the 5 Year Plan

Author Response

Thank you very much for the positive and constructive comments and suggestions on our manuscript. Revised portion are marked in red in the paper and the responds to the comments are as flowing:

Point 1: The introduction would benefit from a wider discussion of the evolution of the livestock sector given that feed demand is such a major force. Perhaps you could start with "China’s livestock transition: Driving forces, impacts, and consequences Zhaohai Bai1,2, Wenqi Ma3, Lin Ma1*, Gerard L. Velthof4, Zhibiao. There is also no mention of climate change which could reduce yield growth.

Response 1: Thank you for recommending the literature. We rewrote the beginning of the introduction based on your positive and constructive suggestions (line 29-33, revised version with track changes). Besides, we added some introduction about the impact of future climate change to grain yield (line 321-323).

Point 2: It would have been good to see some sensitivity analysis regarding key assumptions, e.g.  your assumptions about feed-meat conversions rates are rather pessimistic.

Response 2: The intention of this paper is to make a relatively definite forecast of China's future grain consumption, and then discuss the grain production and security situation, and thus put forward some policy suggestions. Sensitivity analysis is good when discussing the forecasts of grain consumption, but then there will be more uncertainty in the discussion of grain security. In fact, the feed-meat conversion ratios of main livestock products have not been a definite value. The values in this paper are mainly referred to Xin et al.(2015) and Luo et al.(2014), because they investigated and calculated the feed-meat conversion ratios. In this paper, a simple sensitivity analysis about feed-meat conversion ratios was mentioned in the section of policy suggestions in line 357-359. More systematic sensitivity analysis may be the direction of our next study.

Point 3: The discussion of policy options should be related to the actions set out in the 5 Year Plan.

Response 3: In fact, the policy proposed in this paper is consistent with the Five Year Plan. For example, the 13th Five-Year Plan for Agricultural Development calls for “in-depth promotion of cropping structure adjustment…, strengthening the innovation and popularization of science and technology…”. Similarly, based on our study, we propose to adjust the cultivation of maize, expand the cultivation of soybean, and increase the investments in agricultural science and technology.

Reviewer 2 Report

This is an important empirical article that appears to be well crafted. My editorial comments are:

1.Some e=relatively minor English is called for.  The current draft is clear, but sometimes awkward.

2.The prose is sloppy in the srly sections by not being clear between total consumption and per capita consumption.

3. The meaning of a negative standard deviation in Table 3 is unclear.

4.The information in Figure 1 probably does not merit a full page of graphs.

5.In Table 5, the conver4sion ratios are unclear as to whether they refer to live weights, carcass weights, or edible weights.  Generally they look low relative to other estimates, and the beef/mutton number seems especially low.

6.In table 4, why does the model consistently under-predict?

7. The essay def8nes food security implicitly as food self sufficiency.  That is a very poor definition and should be changed

Author Response

Thank you very much for the positive and constructive comments and suggestions on our manuscript. Revised portion are marked in red in the paper and the responds to the comments are as flowing:

Point 1: Some relatively minor English is called for. The current draft is clear, but sometimes awkward.

Response 1: We examined and revised the English writing in this manuscript. The draft is sometime awkward perhaps because some words are easily confused. It needs to be explained that, food grain refers to the grain eaten by residents, feed grain refers to grain that is fed to livestock. Grain’s food, feed and other consumption refers to the different types of grain consumption.

Point 2: The prose is sloppy in the srly sections by not being clear between total consumption and per capita consumption.

Response 2: Total grain consumption is the multiple of per capita grain consumption and population. Sorry for our negligence of showing the changes in the population. We added it in the beginning of the sections of total grain consumption. 

Point 3: The meaning of a negative standard deviation in Table 3 is unclear.

Response 3: Sorry for our mistakes. We re-examined the data and corrected it.

Point 4: The information in Figure 1 probably does not merit a full page of graphs.

Response 4: Figure 1 showed the distribution of inverse roots of characteristic equation of each VAR model. The information the figure reflects is simple, but it is very long, so we put it in the appendix.

Point 5: In Table 5, the conversion ratios are unclear as to whether they refer to live weights, carcass weights, or edible weights.  Generally they look low relative to other estimates, and the beef/mutton number seems especially low.

Response 5: Sorry for our negligence of explaining the definition of feed-meat conversion ratios. Feed-meat conversion ratios refers to the amount of grain consumed in the production of per kilogram of animal products (meat, eggs, milk), they are based on carcass weight. However, the feed-meat conversion ratios of main livestock products have not been a definite value, and they varied greatly in different studies. For example, the minimum conversion rate of pork was 2.5 and the maximum was 4.8; the minimum conversion rate of beef and mutton was 1.0 and the maximum was 4.8; the minimum conversion rate of poultry was 1.5 and the maximum was 2.7; The minimum conversion rate of eggs was 1.6 and the maximum was 2.7; the minimum conversion rate of milk was 0.3 and the maximum was 1.3; the minimum conversion rate of aquatic products was 0.5 and the maximum was 1.5. The values in this paper are mainly referred to Xin et al.(2015) and Luo et al.(2014), because they investigated and calculated the feed-meat conversion ratios. The conversion ratios of beef and mutton were low because that the feeding of cattle and sheep is mainly forage. If the forage is not converted into grain during the calculation, the feed- meat conversion ratio of beef and mutton should be lower.

Point 6: In table 4, why does the model consistently under-predict?

Response 6: In this paper, the model accuracy was calculated as (1-|estimated value-actual value|⁄actual value)*100. We just calculated the absolute value of the simulation accuracy, so the values in Table 4 are all less than 100%. Sorry for our negligence of the formula description, but we annotated it under the table in the revised version.

Point 7: The essay defines food security implicitly as food self-sufficiency.  That is a very poor definition and should be changed.

Response 7: Food security requires three interrelated elements: food availability, access and utilization, and it is not synonymous with food self-sufficiency. But food self-sufficiency ratio is the most important and often used indicator to measure the food security of a country especially for China, because it is of thought that a populous country could only be food-secure if it produces its own food. We added notes in the paper to explain the relationship of grain security and grain self-sufficiency.

Reviewer 3 Report

Comments:

This is a well-written research manuscript that I hope will be eventually worthy of publication in the Sustainability journal. I would like to support the publication of this manuscript. The authors have done certain things well, yet a few revisions and improvements are required. The authors should discuss the main conclusion of the study at the end of the abstract in 1-2 sentences. The Introduction section reads more like magazine report with no critique discussion on the main topic. I still don’t see any research hypotheses in this section. These research hypotheses should be integrated into the text, and be proposed based on current state of knowledge as evidence in the cited literature. This section also could be improved by adding 1-3 research questions at the end; the authors should ensure that all these questions are properly addressed in the Conclusion section. As mentioned before, the Introduction section is very short and weak; the authors should add a few relevant/recent studies along with their approaches and outcomes and indicate the main contribution of the current study by comparing it with previous ones. The Methods section has been improved but it gotten a bit lengthy containing many detailed information. My suggestion is to add a few of tables as appendices at end of the manuscript. The Results section is very short; I suggest merging Results and Discussion sections as a single section. There are many tables in this manuscript; I suggest, if possible, converting a few of tables into figures. The Discussion section needs further enrichment as the discussion of the results according and compared to existent literature is missing. The authors should outline how the main findings are in line with previous studies. I miss more elaboration on the main implications of findings in the Conclusion section. The authors should highlight the future research directions in one paragraph in the Conclusion section as well.

Author Response

ponse 6: We have converted Table 2 into Figure 1.

Point 7: The discussion section needs further enrichment as the discussion of the results according and compared to existent literature is missing. The authors should outline how the main findings are in line with previous studies.

Response 7: Firstly, we added comments about some existing researches. Then we selected three studies with up-to-data and systematic forecast for comparison. By comparison, we found that our findings were in line with these three studies: we all predict that the consumption of food grains (mainly rice and wheat) will not change much, while the consumption of feed grains (mainly maize and soybean) will increase significantly in the following decade.

Point 8: I miss more elaboration on the main implications of findings in the Conclusion section. The authors should highlight the future research directions in one paragraph in the Conclusion section as well.

Response 8: Thank you for the suggestions. We have made revision to the conclusion according to the suggestions.

This manuscript is a resubmission of an earlier submission. The following is a list of the peer review reports and author responses from that submission.